# GENERATIVE MATCHING UNITS FOR SUPERVISED LEARNING

## ABSTRACT

We propose an alternative computational unit for feedforward supervised learning architectures, called Generative Matching Units (GMUs). To understand GMUs, we start with the standard perceptron unit and view it as an undirected symmetric measure of computation between the weights $W = [w_1, w_2, ..w_d]$ and each input datapoint $X = [x_1, x_2, .., x_d]$. Perceptrons forward $W^T X + b$, which is usually followed by an activation function. In contrast, GMUs compute a directed asymmetric measure of computation that estimates the degree of functional dependency $f$ of the input elements $x_i$ of each datapoint to the weights $w_i$ in terms of latent generative variables $\theta$, i.e, $f(w_i, \theta) \rightarrow x_i$. In order to estimate the functional dependency, GMUs measure the minimum error $\sum(f(w_i, \theta) - x_i)^2$ incurred in the generation process by optimizing $\theta$ for each input datapoint. Subsequently, GMUs map the error into a functional dependency measure via an appropriate scalar function, and forward it to the next layer for further computation. In GMUs, the weights $[w_1, w_2, .., w_d]$ can therefore be interpreted as the *generative weights*. We first compare the generalization ability of GMUs and multi-layered-perceptrons (MLPs) via comprehensive synthetic experiments across a range of diverse settings. The most notable finding is that when the input is a sparse linear combination of latent generating variables, GMUs generalize significantly better than MLPs. Subsequently, we evaluate Resnet MLP networks where the first feedforward layer is replaced by GMUs (GMU-MLP) on 30 tabular datasets and find that in most cases, GMU-MLPs generalize better than the MLP baselines. We also compare GMU-MLP to a set of other benchmarks, including TabNet, XGBoost, etc. Lastly, we evaluate GMU-CNNs on three standard vision datasets and find that in all cases they generalize better than the corresponding CNN baselines. We also find that GMU-CNNs are significantly more robust to test-time corruptions.

## 1 INTRODUCTION AND MOTIVATION

We consider the supervised classification problem, where the objective is to predict the output category $y \in \{1, .., c\}$, given the input $X \in \mathbb{R}^d$, and we are given the sample $S = \{(X_1, y_1), .., (X_m, y_m)\} \sim P(X, y)$. Most supervised learning approaches tackle this problem via feedforward architectures. A key building block of most feedforward learners has been the perceptron unit, which involves a linear map $W^T X + b$ followed by some activation function $a(.)$, resulting in the outcome $a(W^T X + b)$, where $X, W, b$ represent the inputs, weights and biases, respectively. This computational unit has been quite ubiquitous across various modern architectures, including transformer variants Lin et al. (2022) and convolutional network variants Li et al. (2021). Even networks which don't specifically have the perceptron as a unit of computation, end up using similar feedforward concepts, such as in Capsule networks Patrick et al. (2022) or in the recently proposed Kolmogorov-Arnold Networks Liu et al. (2024). Multi-layered perceptrons are also proven universal approximators, so given enough complexity and data, they can indeed learn the underlying function to an arbitrary degree of accuracy Hornik et al. (1989).

Mechanistically, if each input dimension of $X$ represents causal variables which impact the output label in a feedforward manner, then the feedforward perceptron unit could potentially end up closely reflecting the actual mechanism involved in yielding the output. This is because MLPs are good function approximators, and thus if the underlying mechanism is indeed a function that maps the

input to the output, they should be preferred. However, what if the underlying mechanism is not feedforward?

Consider scenarios where the actual causes of the data are hidden, and furthermore, where the mechanism of the data generation is related to the output category $y$. More specifically, suppose each datapoint has been generated as $X_i = f(\theta_i, W_{y_i})$, where $\theta_i \in \mathbb{R}^k$, $k < d$, $f : \mathbb{R}^k \rightarrow \mathbb{R}^d$ and $W_{y_i} \in \mathbb{R}^{k \times d}$ represents the *generative weights* corresponding to the ground truth label $y_i$ for $X_i$. $\theta_i$ can be interpreted as the latent generating variables for the instance $X_i$. Note that we will have a total of $c$ generative weights $W_1, W_2, .., W_c$ which corresponding to each output label from $\{1, 2, .., c\}$. The function $f$ represents the structural mechanism via which the hidden variables $\theta_i$ interact with the generative weights $W_{y_i}$. Note that in this scenario, the generative mechanism for the input $X$ is dependent on its ground truth label $y$ and the latent generating variables $\theta$. If the form of $f$ is known, we can construct a network with $c$ output units and corresponding learnable weights $W'_1, .., W'_c$ as follows: for $j = 1, 2, \cdots c$, $v_j = \min_\theta \|X - f(\theta, W'_j)\|$, and the output label estimate is $\hat{y} = \arg\min_{j \in \{1, \cdots c\}} v_j$. As the underlying generative weights $W_1, .., W_c$ are unknown, we would need to optimize $W'_j$ via gradient descent on some loss function, so that the network would ideally converge to the case where the weights $W'_j = W_j$. This represents the intuitive idea behind *generative matching units* (GMU) which we propose, in this paper, as an alternative computational unit for feedforward supervised learning network architectures.

The previous example illustrates an alternative means to create computational units for supervised learning, from the perspective of generative error, using a set of generative weights. This can be beneficial in scenarios where the causal variables that generate the data distribution are hidden, and also directly relate to the ground truth label. Furthermore, for the optimization $\min_\theta \|X - f(\theta, W'_j)\|$ to be fully determinable, it needs to hold that $dim(\theta) < dim(X)$, as otherwise there will be more unknowns than equations. In fact, when the data is high dimensional but exists in low dimensional manifolds ($dim(\theta) << dim(X)$), we can impose the constraint that $k << d$, which imparts a natural dimensionality bias in the unit.

Following the above observations, we can now design a simple GMU as an alternative computational unit. First, we note that it is beneficial to have the GMU output larger values when it is able to better match the input via the optimization $\min_\theta \|X - f(\theta, W'_j)\|$. Thus, we design a unit where smaller $\min_\theta \|X - f(\theta, W'_j)\|$ yield a larger output and vice-versa. Let $X = [x_1, .., x_d] \in \mathbb{R}^d$ be the input, $W = [w_1, .., w_d] \in \mathbb{R}^d$ represent generative weights, $B = [b_1, .., b_d] \in \mathbb{R}^d$ represent the generative bias, $\theta \in \mathbb{R}$ represents the latent generating variable. Then, assuming a linear form for the structural mechanism $f$, the output of the GMU, $G(X)$, is given by:

$$G(X) = \exp\left(-\min_{\theta \in \mathbb{R}} \frac{\sum_{t=1}^{d} (x_t - (w_t\theta + b_t))^2}{d}\right) \tag{1}$$

Note that the above minimization is equivalent to a least-squares problem, and has a well-defined analytical solution. This ensures that $G(X)$ is differentiable w.r.t the weights and a network which uses GMUs in any of its layers can be trained via back-propagation. This leads us to the main motivational points for GMUs, which we summarize below:

1. **Feedforward computation from a generative perspective:** We rethink feedforward computation from a generative standpoint, where each unit attempts to generate the given input datapoint using its associated weights and outputs a function of the error incurred in the generation process. This ensures that our units are primarily computing from a top-down perspective, which is helpful in scenarios where the latent variables that generate the data also relate to the output class.

2. **Input as a function of the weights:** We consider computational units which encode a type of *generative error* in matching the input datapoint. Consider the example of a GMU given in equation 1. Another way to view this relationship is that the generative error encodes whether the input dimensions $x_j$ can be represented as a function of the weights $W_j$ and the biases $b_j$. This is in contrast to the perceptron unit which can be roughly interpreted as the projection of the input on the weights. In our case, the weights may thus be interpreted as the means via which the latent generating variables project onto the input.

3. **Imposing generative complexity constraints:** As most high dimensional data in nature likely has a low dimensional representation due to it existing in low dimensional manifolds, we can

impose complexity constraints on each generative matching unit. An example of such a constraint is simply choosing only a few latent variables in $\theta$, and equation 1 is such an example where $dim(\theta) = 1$.

Even with these ideas, it is worth noting that as classical multi-layered neural networks are universal approximators, they should be able to also learn functions which are of the form depicted in equation 1. However, as we show later on, their ability to interpolate and extrapolate the behaviour of a GMU like $G(X)$ is significantly lower tha networks with GMUs. Similarly, we also show that the ability of a GMU to learn functions which are neural network generated is lower than neural networks. So, via these observations, it is clear that each computational unit has its own inductive biases that enable it to learn and generalize better when the underlying function follows its assumptions. Note that the inductive biases are primarily from a generative standpoint, which in our case is the assumption that there exists local linear generative models that can explain the data. Next, we discuss alternative ways to interpret GMUs, to shed more light on their behaviour.

## 1.1 ALTERNATIVE INTERPRETATIONS OF GMUS

**Directed Measure of Functional Dependence:** There also exists an alternative interpretation of a GMU, and this is one which we initially thought of while thinking of this idea, and it is as follows. First, we realize that the perceptron unit can be considered to be a correlation measure between the input dimensions and the corresponding weights, in certain conditions. Specifically, consider $X \in \mathbb{R}^d$, where $d$ is large and the distance of $X$ from the origin is fixed, i.e. $\|X\| = C_X$. For a trained perceptron with weights $W$, we will have $\|W\| = C_W$. This then yields $W^T X = \left( \frac{W}{\|W\|} \right)^T \frac{X}{\|X\|} \|W\| \|X\| = cos(\theta) C_X C_W$. Here $cos(\theta)$, which represents the cosine of the angle between the two vectors $W$ and $X$, can be interpreted as a normalized measure of correlation between the input dimensions and the corresponding weight values.

We first note that this measure is symmetric, i.e., it remains unchanged when we swap $W$ and $X$. This leads to the question, why not try a more general, and directed measure of dependency between $W$ and $X$? In recent years, many directed functional correlation measures have been proposed of the form $C(X \to Y)$ Chatterjee (2021); Azadkia & Chatterjee (2021), which indicate to what degree the random variable (RV) $Y$ can be represented as a function of $X$. There are also some examples from information theory Xu et al. (2020), where the recently proposed $\mathcal{V}$-Information $I_\mathcal{V}(X \to Y)$ estimates a computationally constrained and directed measure of shared information between $X$ and $Y$. Although some of these measures are hard to compute, and to differentiate (as they are rank based), by considering the general idea, we can also arrive at our proposed GMU structure as follows. As we want to functionally relate the individual dimensions of $X$ denoted by $x_i$ to the corresponding weights $w_i$, we can construct a GMU which is of the form:

$$G(X) = \exp\left( - \min_{\theta \in \mathbb{R}^k} \frac{\sum_{i=1}^d (x_j - f(w_j, \theta))^2}{d} \right), \quad (2)$$

where the function $f$ can come from any function family. For instance, when $f$ is a polynomial of order $k$, we can express $f$ as $f(w_j, \theta) = \sum_{i=1}^k w_j^{i-1} \theta_j$. Later on, we will outline various types of GMUs, and their use cases. We will also discuss various ways in which we can make GMUs behave like functional correlation measures and normalize them, which leads to better training and performance in many cases. Lastly, note that GMUs are asymmetric by definition as they represent a top-down directed measure of functional dependence between the generative weights and the input, as we can see from equation 1 and equation 2.

**A Generalization of RBFs:** We note that radial basis function units are of the form:

$$RBF(X) = \exp\left( - \frac{\sum_{i=1}^d (x_j - c_j)^2}{2\sigma^2} \right), \quad (3)$$

where $c_j$ represents *centers* from which the distance to the current datapoint is taken, and $\sigma$ represents a trainable parameter optimized via gradient descent on an appropriate loss function. Note that when comparing this form with equation 1, the similarities and differences are apparent. First, while RBFs depend on the distance between two $d$ dimensional vectors, GMUs estimate the average

of the per-dimension distance, which avoids scaling up as the dimensionality of the data increases and leads to more stable learning as we show later in this work. For both units, the magnitude of the response depends inversely on the distance. Second, RBF estimates the distance between the input and fixed point centers (which are optimized via gradient descent as well) in $\mathbb{R}^d$, whereas the GMU output in equation 1 depends on the distance between the input datapoint $X$ and a straight line in $\mathbb{R}^d$. Note that the parameters of the straight line are similarly optimized via gradient descent. This shows that in some ways, GMUs can be interpreted as a more flexible generalization of RBFs. It is well known that the distance distribution between two points in $\mathbb{R}^d$ can get significantly uninformative for large $d$, which is also called the *curse of dimensionality* Köppen (2000). We show that distances between points and subspaces, which is a component of GMUs, reduce the impact of this curse. We show in our work, via theory and empirical testing, that GMUs can convey greater information in high-dimensional spaces, when compared to both RBF units and perceptron units, when the data distribution is uniform on a spherical surface.

## 2 CONTRIBUTIONS

This brings us to the overall contributions of our work. We outline them as follows.

- We propose a new computational unit for feedforward supervised learning architectures, called a generative matching unit. We propose several differentiable variants of GMUs and showcase the uses of each form on different types of datasets.

- GMUs consider a generative approach to computation, and we show theoretically and empirically that they can convey more information in high-dimensional spaces, leading to more expressivity. Much smaller GMUs can be highly expressive and learn the underlying function when the low-dimensional generative assumptions are met.

- We show that GMU-based networks and multi-layered perceptrons have their own scenarios where they generalize better than the other. Specifically, in the sparse linear structure prediction problem, and the structure regression problem, we find that GMUs generalize significantly better than MLPs. Similarly, we find that when the ground truth classifier is Naive Bayesian with linear dependencies, MLPs outperform GMUs.

- We conduct an exhaustive comparison of performance between GMU-MLPs and MLPs on 30 tabular datasets. Here, GMU-MLPs refer to the architecture that results after replacing the first layer of the MLP (Resnet) with GMUs. We find that in the majority of cases GMU-MLPs generalize better than MLPs, and in some cases they yield state-of-the-art results, when compared to well-known benchmarks.

- We test GMU-MLP and GMU-CNN variants on three standard vision datasets, focusing on other aspects of performance in addition to test accuracy. Specifically, we find that GMU-MLP and GMU-CNNs significantly outperform their vanilla counterparts in terms of robustness to test-time corruptions.

## 3 GENERATIVE MATCHING UNITS: DEFINITION AND VARIANTS

We first define the most general form of a GMU as follows.

**Definition 1.** *(**Generative Matching Unit:**) Let the input to the unit be $X = [x_1, x_2, .., x_d] \in \mathbb{R}^d$, where every $x_i \in \mathbb{R}$. Consider a function family $\mathcal{F}$, such that every function $f : \mathbb{R}^k \to \mathbb{R}^d \in \mathcal{F}$ can be parameterized as $f(\theta, W) + b$, where $\theta \in \mathbb{R}^k$ $W \in \mathbb{R}^{k \times d}$ $b \in \mathbb{R}^d$. $k$ is denoted as the order of the unit. $W$ and $b$ represent the generative weights and biases of the unit, and $\theta$ represents the latent generating variables. With this, we define the general form of any generative matching unit as follows:*

$$G(X) = \phi \left( \frac{1}{\sigma \sqrt{d}} \min_{\theta \in \mathbb{R}^k} \left\| \frac{X - b}{\eta_X} - f(\theta, W) \right\| \right) \tag{4}$$

*$\phi : \mathbb{R} \to \mathbb{R}$ can be interpreted as an activation function for the GMU. $\eta_X$ represents an optional normalization measure to ensure that $\frac{1}{\sqrt{d}} \min_{\theta \in \mathbb{R}^k} \left\| \frac{X-b}{\eta_X} - (f(\theta, W)) \right\|$ is bounded if needed. Lastly, $\sigma$ represents an optional smoothing factor.*

Table 1: All GMU parameter choices tested in our work

| | Parameter Choices |
|---|---|
| $\theta$ | $var(\mathbb{R}^k)$ |
| $b$ | $var(\mathbb{R}^d)$ ; $\quad [0, 0, .., 0]$ |
| $W$ | $var(\mathbb{R}^{k \times d})$; $\quad [var(\mathbb{R}^{k-1 \times d}); J_{1,d}]$; |
| $f$ | $\theta^T W$ |
| $\phi(z)$ | $e^{-z^2}$; $\quad \sqrt{1 - z^2}$; $\quad -\log z$ |
| $\eta_X$ | $1$; $\quad \sigma_{X-b}$; $\quad \sqrt{\sum_{i=1}^{d} x_i^2 / d}$ |
| $\sigma$ | $1$ |

In our work, to ensure faster computation and differentiation, we only work with functions $f$ such that the minimization $\min_{\theta \in \mathbb{R}^k} \| \frac{X-b}{\eta_X} - f(\theta, W) \|$ can be represented as a least squares problem, which has an analytical solution Wikipedia (2024). Also, note that in this work we set $\sigma = 1$, for reasons that we outline later on. We next outline the GMU variants that we experiment with in this paper. For notational ease, we denote GMU(k) as a GMU of order $k$.

**Remark 1.** *Note that when $k = 0$, a GMU computes $G(X) = \phi\left(\left\| \frac{1}{\sigma\sqrt{d}} \frac{X-b}{\eta_X} \right\|\right)$. When we set $\phi(z) = e^{-z^2}$ and $\eta_X = 1$, this unit becomes similar to an RBF unit with an additional averaging factor $d$ that averages the distance across all dimensions.*

### 3.1 GMU VARIANTS

We outline all parameter and function choices in equation 4 tested in this paper in Table 1. Note that in Table 1, all real number based entries of the form $var(\mathbb{R}^{p \times q})$ denote tensors of size $p \times q$ where all entries are real variables subject to gradient descent. Lastly, $J_{p,q}$ represents a fixed matrix of all ones of size $p \times q$.

**Remark 2.** *Note that we only consider linear $f$ in our work, which enables us to formulate $\min_{\theta \in \mathbb{R}^k} \left\| \frac{X-b}{\eta_X} - f(\theta, W) \right\|^2 = \min_{\theta \in \mathbb{R}^k} \left\| \frac{X-b}{\eta_X} - \theta^T W \right\|^2$ as a linear least squares problem. Linear least-squares has a fixed analytical solution Wikipedia (2024), which then enables quick computation and also differentiation from a gradient descent based optimization perspective. We therefore denote the variants explored in our work as Linear GMUs. Geometrically, the GMU output can be interpreted as a distance measure between the input point $X$ and the linear subspace which is modelled by $\theta^X$. We showcase this geometric property in Figure 2 (a).*

### 3.2 ARE GMUS UNIVERSAL APPROXIMATORS?

A single GMU clearly cannot act as a universal approximator, as it can only model functions of the form in equation 4. We study linear GMU-MLPs, which consists of a layer containing multiple linear GMUs followed by a perceptron unit. We study whether GMU-MLPs can be universal approximators. Note that it is well known that RBFs are universal approximators, and as our GMU with $k = 0$ has a similar form, it is likely that it would be a universal approximator as well. First, let us define the set of functions $L^p(\mathbb{R}^d)$ such that any $f \in L^p(\mathbb{R}^d)$, where $f : \mathbb{R}^d \to \mathbb{R}$, is $p^{th}$ power integrable, bounded, continuous and continuous with compact support. $L^p(\mathbb{R}^d)$ encompasses the set of all such functions which satisfy these constraints. This leads to our first result.

**Proposition 1.** *(from Park & Sandberg (1991)) A linear GMU-MLP with $k = 0, \eta_X = 1$ and $\phi(z)$ being any integrable bounded function such that $\int \phi(x)dx \neq 0$ can approximate any function $f \in L^p(\mathbb{R}^d)$.*

We found that showing linear GMU with $k > 0, \eta_X = 1$ are universal approximators for any function $f \in L^p(\mathbb{R}^d)$ is non-trivial. We instead provide an intuitive geometrical argument to support the hypothesis that Linear GMU-MLPs of any order $k$ are universal approximators. which finds that multiple linear GMUs with $k > 0$ can be used to approximate the behaviour of a linear GMU with $k = 0$. We provide our argument as follows, for the case of $k = 2$ and $d = 3$, which can be extended to other cases similarly.

First, let $X = (x, y, z)$ and let us consider a linear GMU with $k = 0$, which computes $G(X) = \exp{-\frac{1}{3}\left((x-a)^2 + (y-b)^2 + (z-c)^2\right)}$. We can construct three linear GMUs with $k = 2$ in the manner shown in Figure 2 (b) in the Appendix, where the subspaces (2D planes) are chosen such that $G_1(X) = \exp{-(x-a)^2}$, $G_2(X) = \exp{-(y-b)^2}$ and $G_3(X) = \exp{-(z-c)^2}$. As we are considering GMU-MLPs, we can average the units in the next layer, to yield: $\frac{1}{3}\left(G_1(X) + G_2(X) + G_3(X)\right) = \exp{-(x-a)^2} + \exp{-(y-b)^2} + \exp{-(z-c)^2}$. Note that this unit behaves similar to the original linear GMU $G(X)$, in that it attains its maximum value when $x = a, y = b, z = c$, like $G(X)$. Also, points closer to $(a, b, c)$ are likely to yield larger activations than the ones farther away for the averaged unit, similar to $G(X)$. Lastly, when $\|(x, y, z) - (a, b, c)\| \to 0$, we can approximate $\frac{1}{3}\left(G_1(X) + G_2(X) + G_3(X)\right) \approx \exp{-\frac{1}{3}\left((x-a)^2 + (y-b)^2 + (z-c)^2\right)} = G(X)$. Subsequently, the second layer weights associated with $G_i(X)$ can all be set to $W/3$ to yield the same function as the GMU-MLP with $k = 0$ and weights $W$. This argument can be extended to arbitrary $k, d$ in the same manner. This shows that linear GMU-MLPs with $k > 0$ can potentially approximate GMU-MLPs with $k = 0$, but using more hidden units. As GMU-MLPs with $k = 0$ are universal approximators (Proposition 2), this implies that GMU-MLPs with arbitrary $k$ can potentially be universal approximators as well.

## 4 GMUs and the Curse of Dimensionality

We first define the notion of *information factor*, which represents the normalized variability of any similarity measure $S(X_1, X_2)$ in $d$ dimensional space, where $X_1, X_2 \in \mathbb{R}^d$. We only consider distance measures in this analysis.

**Definition 2.** *(Information Factor) We are given a similarity measure $S(X_1, X_2)$, where $X_1, X_2 \in \mathbb{R}^d$. Let $I_d$ represent the identity matrix of size $d \times d$. Then, information factor $\gamma_S(d)$ of $S$ in $d$-dimensional space is estimated as:*

$$\gamma_S(d) = \frac{\mathbb{E}_{X_1, X_2 \sim \mathcal{N}(0, I_d)}\left[\sigma\left((S(X_1, X_2))\right]}{\mathbb{E}_{X_1, X_2 \sim \mathcal{N}(0, I_d)}\left[\mu\left((S(X_1, X_2))\right]}, \tag{5}$$

*where $\sigma(x)$ and $\mu(x)$ denote the standard deviation and the mean of $x$ respectively.*

With this, we undergo a series of experiments where we estimate the information factor of multiple measures, including Euclidean distance and the distance measured in GMUs, which is the distance between a linear subspace of dimensionality $k$ and the input point. We denote this as the $k$-subspace distance. But before that, we first provide some theoretical results that compare the information factor of Euclidean distance and $k$-subspace distances

**Theoretical Results:** We outline our first result for Euclidean distances as follows:

**Proposition 2.** *Let $E(X_1, X_2) = \|X_1 - X_2\|$, we can show that $\gamma_E(d+1) < \gamma_E(d)$.*

Next, we outline the analogous result for $k$-subspace distances.

**Proposition 3.** *We define the $k$-subspace distance from $X_1$ to $X_2$ as $S_{k,W}(X_1, X_2) = \min_{\theta \in \mathbb{R}^k}\|X_2 - (\theta^T W + X_1)\|$ With this, first, we note that $\gamma_{S_{0,W}}(d) = \gamma_E(d)$, where $E$ denotes the Euclidean distance. Then, we have that $\gamma_{S_{k+1,W}}(d) > \gamma_{S_{k,W}}(d)$, and thus $\gamma_{S_{k,W}}(d) > \gamma_E(d)$.*

**Remark 3.** *Propositions 3 and 2 highlight a few interesting points. First, we see that the information factor of Euclidean distance decreases with dimensionality, which is another way to interpret the curse of dimensionality. Variation in distance reduces in high dimensional spaces, leading to loss of structure and thus making it harder for distance based approaches such as nearest-neighbor or RBFs to work with the data. However, as proposition 3 shows, the information factor of the $k$-subspace distance, which is measured in GMUs, is strictly larger than of Euclidean distance. Furthermore, we see that the information factor increases as the order of the GMU, $k$, increases. This shows that high-order GMUs may be helpful in extracting more structural information in high-dimensional datasets, than just Euclidean distance.*

**Empirical verification:** We verify the results in Propositions 3 and 2 by simulating $X_1, X_2 \sim \mathcal{N}(0, I_d)$ and estimating the information factor for the Euclidean distance $\gamma_E(d)$ and the $k$-subspace distance $\gamma_{S_{k,W}}(d)$, as a function of the dimensionality $d$. We summarize all our empirical findings in Figure 1.

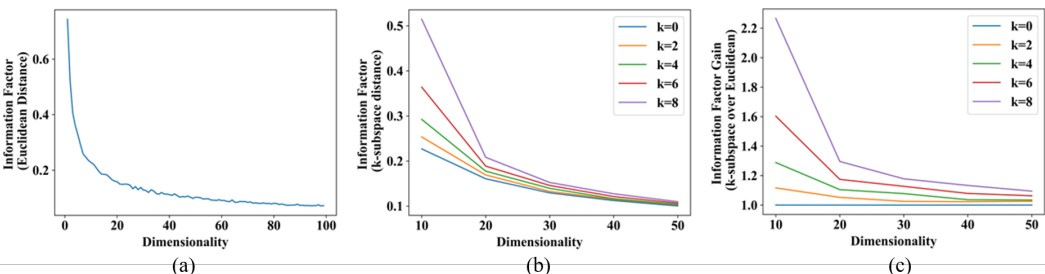

Figure 1: Information Factor v/s Dimensionality: (a) For Euclidean Distance (b) for $k$-subspace Distance and (c) Information Factor Gain ($\gamma_{S_{k,W}}(d)/\gamma_E(d)$).

**Takeaways:** We find that our observations agree with the propositions. Specifically, we see that the Euclidean distance information factor first shows a significant decrease with increase in data dimensionality, reaching values very close to 0. The same is observed for the $k$-subspace distance, however, we see that as $k$ increases, the information factor increases. Figure 1 (c) shows the gain in the information factor ($\gamma_{S_{k,W}}(d)/\gamma_E(d)$) as a function of $d$ and $k$. This plot shows a clear improvement in the information factor when using $k$-subspace distance as opposed to Euclidean distance.

## 5 SYNTHETIC EXPERIMENTS

In this section, we conduct experiments where the data distribution is artificially generated. Each experiment showcases a different type of distribution. We discuss the takeaways from these experiments at the end of this section.

### 5.1 SPARSE LINEAR STRUCTURE PREDICTION

**Problem Outline:** We argue that most sampled data in nature have a sparse set of causes (latent generating variables) that are active in each instance. This concept has already been studied in sparse representation learning Lee et al. (2006), however, we create a scenario where the set of active causes corresponding to any instance also indicates the underlying output label of that instance. In this way, given a total of $N_c$ latent generating variables, the set of active variables for each instance can be construed as a *sparse linear structure* for the ground truth label of that instance. We formally outline the sampling process as follows.

**Definition 3.** *(Sparse Linear Structure Sampling:) We are given the input RV $X \sim \mathbb{R}^d$ and the ground truth label RV $y \sim \{1, 2.., N_y\}$, such that $(X, y) \sim P(X, y)$. $W \sim \mathbb{R}^{N_c \times d}$ represent the set of generative weights for the $N_c$ latent variables $\{\theta_1, \theta_2, ..\theta_{N_c}\}$. Lastly, for each label $y$, let the set of active latent variables for each instance corresponding to that label be denoted by $\theta_{l_y(1)}, \theta_{l_y(2)}..., \theta_{l_y(y_c)}$, $1 \le y_c \le N_{max}$, where $N_{max}$ denotes the maximum number of generating variables active for any instance. With this, we can outline the generative process as follows. First we sample $y \sim Unif\{1, 2, .., N_y\}$, and then we sample an instance for that $y$ as $x(y) = \sum_{i=1}^{y_c} \theta_{l_y(i)} W_{l_y(i),*} + \epsilon$, where $\epsilon \sim \mathcal{N}(0, \sigma^2 I_d)$.*

**Experiments:** We conduct a series of experiments with different choices of parameters involved in the sparse linear structure sampling process. The elements of the generative weights $W$ are sampled according to the standard normal distribution $\mathcal{N}(0, 1)$, and $y_c$ is chosen uniformly at random from the valid range. The number of training data samples is fixed at 1000 and the number of test data samples at 3000. We compare a two layer MLP with 512 hidden units (relu-activated), a Resnet with 2 groups with 1 block per group and 512 units in each layer, and a Resnet with 2 groups with 2 blocks per group and 512 units in each layer. We denote them as MLP-512, Resnet-512-[2,1], Resnet-512-[2,2] respectively (abbreviated in the Table for space). We provide the results for Resnet-512-[2,2] variants in the Supplementary Materials. We compare these baselines with a single GMU layer consisting of $d$ inputs and $N_y$ outputs, denoted as GMU($k$), where $k$ is its order. The results are summarized in Table 2. Note that the 'Test Config' column represents whether the test data is generated out-of-distribution, i.e., whether the range of the latent generating variables $\theta_{l_y(i)}$

| Setup | | | | | Test Config | GMU(1) | GMU(2) | GMU(3) | GMU(4) | GMU(5) | GMU(6) | MLP | MLP (norm.) | R-[1,1] | R-[1,1] (norm.) |
|---|---|---|---|---|---|---|---|---|---|---|---|---|---|---|---|
| $N_y$ | $N_c$ | $d$ | $\sigma$ | $N_{max}$ | | | | | | | | | | | |
| 20 | 25 | 10 | 0.01 | 3 | same | 0.8997 | 0.9311 | 0.9654 | 0.9794 | 0.9694 | 0.966 | 0.9505 | 0.9728 | 0.9622 | 0.9714 |
| 20 | 25 | 10 | 0.01 | 3 | ood | 0.685 | 0.84525 | 0.9735 | 0.8922 | 0.928 | 0.9442 | 0.5725 | 0.7087 | 0.631 | 0.6525 |
| 20 | 25 | 100 | 0.1 | 3 | same | 0.944 | 0.9271 | 0.9348 | 0.9488 | 0.9482 | 0.9482 | 0.9197 | 0.9451 | 0.946 | 0.9462 |
| 20 | 25 | 100 | 0.1 | 3 | ood | 0.8785 | 0.9005 | 0.926 | 0.848 | 0.8592 | 0.9092 | 0.6457 | 0.733 | 0.681 | 0.724 |
| 20 | 25 | 100 | 0.01 | 3 | same | 0.9471 | 0.9494 | 0.9814 | 0.9874 | 0.9862 | 0.9825 | 0.9545 | 0.9754 | 0.9654 | 0.9788 |
| 20 | 25 | 100 | 0.01 | 3 | ood | 0.892 | 0.9215 | 0.981 | 0.9335 | 0.9365 | 0.9612 | 0.6282 | 0.7492 | 0.693 | 0.7522 |
| 20 | 25 | 100 | 0 | 3 | same | 0.9474 | 0.9565 | 0.9962 | 0.9951 | 0.9974 | 0.9991 | 0.9548 | 0.9797 | 0.9622 | 0.9868 |
| 20 | 25 | 100 | 0 | 3 | ood | 0.892 | 0.925 | 0.9977 | 0.9915 | 0.9835 | 0.9905 | 0.6275 | 0.7495 | 0.6842 | 0.7432 |
| 20 | 25 | 500 | 0.01 | 3 | same | 0.9502 | 0.9434 | 0.9831 | 0.9891 | 0.9894 | 0.9874 | 0.9537 | 0.9771 | 0.9685 | 0.9845 |
| 20 | 25 | 500 | 0.01 | 3 | ood | 0.9087 | 0.9215 | 0.9845 | 0.9395 | 0.948 | 0.9655 | 0.59725 | 0.7677 | 0.6467 | 0.7665 |
| 20 | 25 | 1000 | 0.01 | 3 | same | 0.9111 | 0.9502 | 0.9834 | 0.9957 | 0.9914 | 0.996 | 0.9662 | 0.9814 | 0.9762 | 0.9845 |
| 20 | 25 | 1000 | 0.01 | 3 | ood | 0.8657 | 0.9017 | 0.975 | 0.9537 | 0.987 | 0.983 | 0.517 | 0.7525 | 0.6572 | 0.7837 |
| 20 | 25 | 1000 | 0.01 | 6 | same | 0.978 | 0.9385 | 0.9791 | 0.9908 | 0.9885 | 0.9951 | 0.9637 | 0.9882 | 0.9757 | 0.9882 |
| 20 | 25 | 1000 | 0.01 | 6 | ood | 0.9272 | 0.6867 | 0.9462 | 0.9462 | 0.9905 | 0.9997 | 0.6407 | 0.9107 | 0.641 | 0.904 |
| 50 | 10 | 1000 | 0.01 | 3 | ood | 0.5752 | 0.743 | 0.8997 | 0.9077 | 0.891 | 0.8712 | 0.5492 | 0.5575 | 0.4132 | 0.5352 |

Table 2: Test accuracy results on the sparse linear structure prediction experiments.

| Setup ($N_y = 10$, $\sigma_0 = 0.1$) | | | | GMU(0) -MLP | GMU(1) -MLP | GMU(2) -MLP | GMU(3) -MLP | GMU(4) -MLP | GMU(5) -MLP | GMU(6) -MLP | GMU(7) -MLP | GMU(8) -MLP | MLP | R-[1,1] | R-[2,2] |
|---|---|---|---|---|---|---|---|---|---|---|---|---|---|---|---|
| $d$ | $\Delta$ | Train | Test | | | | | | | | | | | | |
| 10 | 2 | G | G | 0.5548 | 0.564 | 0.5608 | 0.5574 | 0.5608 | 0.5597 | 0.5605 | 0.5568 | 0.5571 | 0.576 | 0.5188 | 0.502 |
| 10 | 4 | G | G | 0.4362 | 0.4371 | 0.4277 | 0.4248 | 0.4265 | 0.4331 | 0.4342 | 0.44 | 0.4394 | 0.4345 | 0.3657 | 0.3525 |
| 50 | 4 | G | G | 0.6943 | 0.9286 | 0.9371 | 0.9420 | 0.9394 | 0.9403 | 0.9406 | 0.9394 | 0.9440 | 0.7965 | 0.8128 | 0.758 |
| 50 | 8 | G | G | 0.6403 | 0.8471 | 0.8686 | 0.8617 | 0.8649 | 0.8680 | 0.8651 | 0.8714 | 0.8686 | 0.7048 | 0.7194 | 0.6802 |
| 100 | 8 | G | G | 0.4503 | 0.9623 | 0.9703 | 0.9769 | 0.9769 | 0.9806 | 0.9797 | 0.9797 | 0.9783 | 0.7617 | 0.7931 | 0.7568 |
| 500 | 8 | G | G | 0.2371 | 0.9991 | 0.9991 | 0.9983 | 0.9980 | 0.9989 | 0.9997 | 0.9994 | 1.0000 | 0.7437 | 0.8102 | 0.772 |
| 500 | 8 | GS | G | 0.3980 | 0.7154 | 0.7829 | 0.5966 | 0.7009 | 0.7806 | 0.7777 | 0.8477 | 0.8917 | 0.4157 | 0.4122 | 0.3448 |
| 500 | 8 | G | GM1 | 0.1583 | 0.7760 | 0.8054 | 0.7783 | 0.7883 | 0.7463 | 0.7963 | 0.8374 | 0.8374 | 0.29 | 0.2691 | 0.2317 |
| 500 | 8 | G | GM2 | 0.2237 | 0.9929 | 0.9960 | 0.9954 | 0.9960 | 0.9943 | 0.9954 | 0.9971 | 0.9983 | 0.6531 | 0.6974 | 0.6551 |
| 500 | 8 | MG | MGN | 0.2071 | 0.9657 | 0.9586 | 0.9697 | 0.9551 | 0.9771 | 0.9789 | 0.9740 | 0.9803 | 0.6394 | 0.6471 | 0.6 |

Table 3: Test accuracy results on the dynamic tree structure prediction experiments.

at test-time is different from the corresponding range at training time. More details are provided in the appendix.

## 5.2 DYNAMIC TREE STRUCTURE PREDICTION

**Problem Outline:** In this section, we consider the setting when the underlying generative model of the data does not depend on unseen latent variables as before, but is present within the data dimensions itself. More specifically, we assume tree based generative models, where every node of the tree corresponds to a specific data dimension. We outline the generative process as follows.

**Definition 4.** *(Dynamic Tree Structure Sampling) We are given $X = [x_1, x_2, .., x_d] \in \mathbb{R}^d$ and $y \in \{1, 2, .., N_y\}$. We have $N_y$ trees denoted as the undirected graphs $\{G_1, G_2, .., G_{N_y}\}$ where $G_i = \{V_i, E_i\}$ and $|V_i| = d \ \forall i$. The vertices of every $G_i$ correspond to the dimensions of $X$, and all trees have the same maximum degree $\Delta$. Define $\alpha \in \mathbb{R}^d$. We outline the generative process as follows. First, we sample $y \sim Unif\{1, 2, ..., N_y\}$ and then we sample a datapoint by tracing the graph $G_y$ as: $x_i = \alpha_i x_{pa(i, G_y)} + \epsilon_i$, where all $\epsilon_i$ are randomly generated for each $x_i$ according to some fixed distribution $P_\epsilon$, and $pa(i, G_y)$ denotes the parent of $x_i$ considering the tree $G_y$. The root node of $G_y$ denoted as $r(G_y)$ is sampled as $x_{r(G_y)} \sim \mathcal{N}(0, \sigma_0^2)$.*

**Experiments:** We pick a range of parameter choices for the sampling process, and compare the following networks: GMU($k$)-MLP-512 (GMU layer with 512 hidden units followed by a linear layer), MLP-512, Resnet-512-[2,1] and Resnet-512-[2,2]. Note that $\sigma_0 = 0.1$ and $N_y = 10$ is fixed for all cases. The tree graphs $\{G_1, G_2, .., G_{N_y}\}$ are created randomly, and each one is assigned to the corresponding class in $y$ for each run. Unless otherwise specified, we set $\alpha_i = 1$. The results are shown in Table 3. For the Table abbreviations, G: Gaussian $\epsilon_i$, GS: Skewed Gaussian $\epsilon_i$ (Shape parameter 4), GM1: Gaussian $\epsilon_i$ and $\alpha_i \sim \mathcal{N}(0, 1)$, GM2: Gaussian $\epsilon_i$ and $\alpha_i \sim \mathcal{N}(0, 4)$ (randomly sampled each time), MG: Gaussian $\epsilon_i$, $\alpha_i \sim mathcalN(0, 1)$ (Fixed), MGN: Gaussian $\epsilon_i$, $\alpha_i \alpha_i^{train} \sim \mathcal{N}(0, 1)$ where $\alpha_i^{train}$ is the $\alpha_i$ set at training time.

## 5.3 PREDICTION ON POLYNOMIAL NAIVE BAYES SAMPLED DATA

**Problem outline:** We consider the scenario where the ground truth labels $y$ are generated in a naive Bayesian manner as $P(y|X) \propto \Pi_i Q(y|X_i)$, where the distribution $Q(x)$ is of the form $e^{-G(x)}/Z$. We formally outline the sampling process as follows.

| Setup $p$ | GMU(0) | GMU(1) | GMU(2) | GMU(3) | GMU(0)-MLP | GMU(1)-MLP | GMU(2)-MLP | GMU(3)-MLP | Linear | MLP |
|---|---|---|---|---|---|---|---|---|---|---|
| 1 | 0.8223 | 0.8406 | 0.8583 | 0.8643 | 0.9843 | 0.9763 | 0.9743 | 0.97 | 0.983 | 0.9823 |
| 2 | 0.9363 | 0.9363 | 0.9363 | 0.9363 | 0.9816 | 0.9873 | 0.986 | 0.982 | 0.9763 | 0.9801 |
| 3 | 0.7916 | 0.803 | 0.816 | 0.8336 | 0.9596 | 0.975 | 0.974 | 0.9746 | 0.9456 | 0.9606 |

Table 4: Test accuracy results on the polynomial naive Bayes sampling experiments.

| Setup | | | GMU(0)-MLP | GMU(1)-MLP | GMU(2)-MLP | GMU(3)-MLP | Linear | MLP | R-[1,1] | R-[2,2] |
|---|---|---|---|---|---|---|---|---|---|---|
| $N_y$ | $d$ | $\sigma_\mu$ | | | | | | | | |
| 2 | 10 | 0.01 | 0.9942 | 0.9888 | 0.9914 | 0.9825 | 0.6474 | 0.9848 | 0.9817 | 0.9782 |
| 10 | 10 | 0.01 | 0.2714 | 0.2608 | 0.244 | 0.2405 | 0.1262 | 0.1908 | 0.1582 | 0.1462 |
| 10 | 10 | 0.1 | 0.8888 | 0.8511 | 0.8254 | 0.8091 | 0.3862 | 0.7685 | 0.7585 | 0.7511 |
| 2 | 100 | 0.01 | 0.7151 | 0.704 | 0.6902 | 0.6851 | 0.5577 | 0.6077 | 0.5694 | 0.5594 |
| 2 | 100 | 0.1 | 0.9414 | 0.9428 | 0.9462 | 0.946 | 0.782 | 0.9248 | 0.9031 | 0.89 |
| 10 | 100 | 0.1 | 0.5145 | 0.4985 | 0.4897 | 0.4897 | 0.2234 | 0.4068 | 0.3214 | 0.2674 |
| 2 | 500 | 0.1 | 0.7377 | 0.7411 | 0.7388 | 0.7454 | 0.6751 | 0.7285 | 0.6905 | 0.6848 |
| 2 | 500 | 1 | 0.986 | 0.9911 | 0.99514 | 0.9951 | 0.9891 | 0.9928 | 0.972 | 0.9794 |

Table 5: Test accuracy results on the class-conditioned Gaussian experiments.

**Definition 5.** *(Polynomial Naive Bayes Sampling) We are given $X \in \mathbb{R}^d \sim Unif(0,1)^d$, and $y \in \{1, 2, 3, ..N_y\}$. We consider a naive Bayesian sampling of $P(y|X) = \frac{1}{Z}\Pi_i P(y|X_i)$, where $P(y|X_i) = e^{-G(y|X_i)}$. Let the polynomial order of the sampling be denoted as $p$. Define a set of weight matrices $\{W_1, W_2, .., W_d\}$, where $W_i \in \mathbb{R}^{N_y \times p}$. We consider $G(y_i|X_j) = W_{ji}[X_j, X_j^2, .., X_j^p]^T$. With this, we outline the sampling as follows. First, we sample $X \sim Unif(0,1)^d$, and then we compute*

$$y* = \underset{i \in \{1,2,3,..N_y\}}{\arg\max} \Pi_i \log P(y = i|X) = \underset{i \in \{1,2,3,..N_y\}}{\arg\max} \sum_{j=1}^d W_{ji}[X_j, X_j^2, .., X_j^p]^T, \quad (6)$$

*where $y*$ denotes the output label for the sampled $X$.*

**Remark 4.** *Note that the expression $\sum_{j=1}^d W_{ji}[X_j, X_j^2, .., X_j^p]^T$ can also be interpreted as the logits in the sampling process, as when $p = 1$, they are simply linear functions of the input, and thus should be solvable via a single linear layer followed by a softmax operator.*

**Experiments**: Every element of all $W_i$ matrices are sampled randomly from $\mathcal{N}(0,1)$. We only vary the polynomial order $p$. $d$ is fixed at 10. We compare the following networks: GMU($k$), GMU($k$)-MLP-512 (GMU layer with 512 hidden units followed by a linear layer), single Linear layer and MLP-512. Results are shown in Table 4.

## 5.4 PREDICTION ON CLASS-CONDITIONED GAUSSIAN DISTRIBUTIONS

**Problem outline:** We conduct a simple experiment where the conditional distributions $P(X|y)$ are Gaussian, where $X \in \mathbb{R}^d$ and $y \in \{1, 2, .., N_y\}$. Specifically, we generate $P(X|y) \sim \mathcal{N}(\mu_y, \Sigma)$. For each dataset, we choose the class-wise mean values by randomly generating them as $\mu_y \sim \mathcal{N}(0, \sigma_\mu^2 I_d)$. Similarly, we pick a randomly generated covariance matrix via $\Sigma = A^T A$ where $A \sim Unif(0,1)^{d \times d}$.

**Experiments:** We pick a range of parameter choices for the sampling process, and compare the following networks: GMU($k$)-MLP-512 (GMU layer with 512 hidden units followed by a linear layer), MLP-512, Resnet-512-[2,1] and Resnet-512-[2,2]. The results are shown in Table 5.

## 5.5 OVERALL TAKEAWAYS

Overall, we find that the GMU variants show significantly better generalization, especially to out-of-distribution test data. Also, we observe that in the cases where the inputs are structured via a common causal framework, such as the sparse and tree structure prediction experiments, more input dimensionality becomes a blessing rather than a curse. This is simply because of the law of large numbers. When high dimensional inputs all share the same cause, with the correct assumptions one can obtain a more accurate estimate of the underlying generating variables, unlike the low-dimensional case. Therefore, it is notable that we see larger dimensionality help performance in the structure prediction experiments, while doing the opposite in the Gaussian experiment.

| Dataset | Resnet | GMU-Resnet | Dataset | Resnet | GMU-Resnet | Dataset | Resnet | GMU-Resnet | Dataset | Resnet | GMU-Resnet | Dataset | Resnet | GMU-Resnet |
|---|---|---|---|---|---|---|---|---|---|---|---|---|---|---|
| anneal | 0.8525 | **0.861** | phoneme | **0.8940** | 0.8882 | jasmine | 0.7419 | **0.7520** | jungle | 0.9615 | **0.9773** | miniboone | 0.8322 | **0.9048** |
| kr-vs-kp | **0.9969** | **0.9969** | cnae | 0.9259 | **0.9398** | sylvine | 0.9161 | **0.9268** | volkert | 0.6796 | **0.7003** | walking | 0.6246 | **0.6322** |
| mfeat | 0.9750 | **0.9800** | blood | 0.6265 | **0.6718** | adult | 0.7717 | **0.7735** | helena | **0.2207** | 0.2206 | ldpa | **0.6980** | 0.6777 |
| credit | **0.7298** | 0.7036 | australian | 0.8596 | **0.8726** | nomao | 0.9591 | **0.9599** | connect | 0.7346 | **0.7535** | aloi | 0.9666 | **0.9684** |
| vehicle | 0.8266 | **0.8793** | car | **1.0** | **1.0** | bank | 0.7371 | **0.7382** | higgs | 0.6718 | **0.6781** | skin-seg | **0.9997** | 0.9996 |
| kc1 | 0.6789 | **0.6866** | segment | 0.9221 | **0.9307** | shuttle | **0.9870** | 0.9823 | numerai | 0.5045 | **0.5146** | arrhythmia | 0.2918 | **0.3259** |

Table 6: Balanced Accuracy on 30 Tabular datasets from OpenML.

| | | | | | Dataset: MNIST | | | | | | |
|---|---|---|---|---|---|---|---|---|---|---|---|
| Network | Standard | brightness | canny | dotted | fog | glass | identity | impulse | motion | shot | spatter | zigzag |
| CNN | 0.9949 | 0.2274 | 0.6149 | 0.9791 | 0.1188 | 0.541 | 0.9949 | 0.4529 | 0.9675 | 0.9226 | 0.9834 | 0.7826 |
| GMU-CNN | 0.9954 | 0.9913 | 0.8998 | 0.9896 | 0.9234 | 0.8141 | 0.9956 | 0.9377 | 0.9614 | 0.9449 | 0.9759 | 0.9447 |
| | | | | | Dataset: Fashion-MNIST | | | | | | |
| CNN | 0.9329 | 0.4535 | 0.3709 | 0.8786 | 0.2712 | 0.6518 | 0.9329 | 0.2056 | 0.7188 | 0.5959 | 0.8835 | 0.8131 |
| GMU-CNN | 0.9356 | 0.8250 | 0.7058 | 0.9118 | 0.7442 | 0.5817 | 0.9298 | 0.6703 | 0.6831 | 0.48964 | 0.8868 | 0.8806 |

Table 7: Test Accuracy of networks trained on MNIST and Fashion-MNIST.

# 6 EXPERIMENTS ON REAL DATASETS:

## 6.1 TABULAR DATASETS

**Outline:** We test and compare performance on 30 datasets from openML. Specifically, we test on a subset of the datasets tested in Kadra et al. (2021). For the Resnet-512-[1,1] architecture, which performed well across our synthetic experiments overall, we replace the first layer with four types of GMU units: $k = 0, 1, 2, 3$. For each $k$, we thus have 128 units in the first layer, yielding a total of 512 output units for the first layer, same as the Resnet. We denote this network as the GMU-Resnet-512-[1,1] architecture.

**Takeaways:** We find that overall, in 25 out of 30 cases, GMU-Resnet-512-[1,1] showcases better or on-par balanced accuracy. Furthermore, the GMU-Resnet-512-[1,1] performs favorably against most other approaches in Kadra et al. (2021) (excluding MLP+C and MLP+Dropout) when compared one-to-one via the Wins/Losses/Ties criterion. Our results re-inforce the observation that while GMU units can impart a better inductive bias in many cases, it can also suffer in other scenarios.

## 6.2 VISION DATASETS

We construct convolutional GMUs, using which we create GMU-CNN architectures, where the first layer is a convolutional GMU and the rest of the network comprises traditional convolutional layers followed by a fully connected layer. We mainly focus on out-of-distribution generalization, by introducing unseen corruptions at test-time, to see if the GMU-CNN recognizes the concept of the vision classes better overall. We report our findings across three datasets: MNIST, Fashion-MNIST, CIFAR-10 and their corrupted versions. Results are summarized in Table 7. We find that GMU-CNNs show significant improvements to test-time corruptions. In general, we observe that trained GMU-CNNs are naturally more robust to distribution shifts.

# 7 CONCLUSION

Our work demonstrates the potential advantages of an alternative computational unit that computes from a generative perspective, imposing a low-complexity constraint on the generation process. Generative Matching Units showcase better generalization and demonstrates a significantly higher ability to identify dynamic causal structures in the inputs. On real tabular datasets, Resnets replaced with GMU layers in the first layer show signifcant performance improvements. Many possibilities remain open for incorporating GMUs in larger networks across other domains, and also find ways to cascade multiple GMU layers.

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

# Supplementary Materials

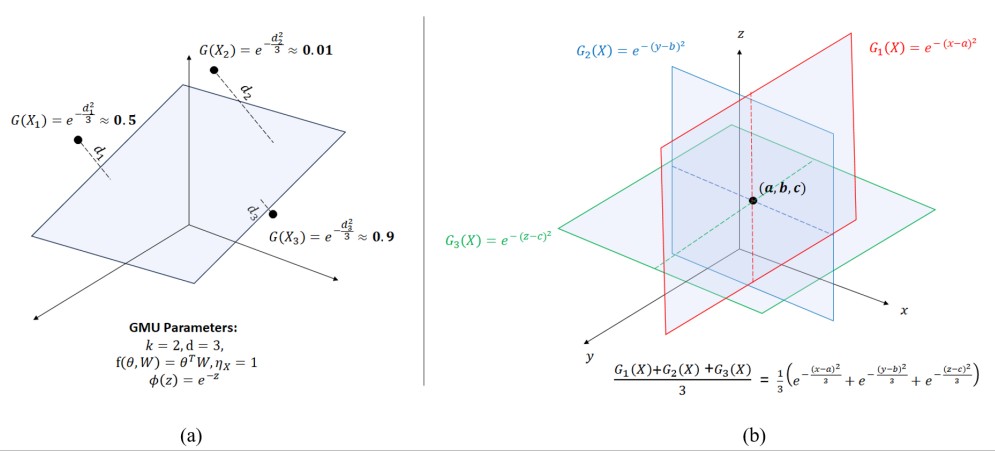

Figure 2: (a) shows an example of a GMU with non-zero bias, no normalization, $\phi(z) = e^{-z^2}$ and $f(\theta, W) = \theta^T W$ and (b) highlights our argument that higher order GMUs can be constructed to mimic a GMU of order zero (RBF).

## A    THEORETICAL PROOFS

**Proposition 4.** *(From Park & Sandberg (1991)) A linear GMU-MLP with $k = 0, \eta_X = 1$ and $\phi(z)$ being any integrable bounded function such that $\int \phi(x)dx \neq 0$ can approximate any function $f \in L^p(\mathbb{R}^d)$.*

*Proof.* We note that when we set $\sigma'^2 = \sigma^2 d$, $k = 0$ and $\eta_X = 1$ the GMU unit essentially becomes an RBF unit. As $\sigma$ here is the same across all hidden units, it implies $\sigma'$ is the same as well, and the results from Theorem 1 in Park & Sandberg (1991) apply. This completes the proof. □

**Proposition 5.** *Let $E(X_1, X_2) = \|X_1 - X_2\|$, we can show that $\gamma_E(d+1) < \gamma_E(d)$.*

*Proof.* Using the statistics of the chi-squared distribution, it is trivial to show that $\gamma_E(d) = \frac{d\Gamma(\frac{d}{2})^2}{\Gamma(\frac{d+1}{2})^2} - 1$. This ultimately leads to the fact that we need to show that

$$\frac{\Gamma(\frac{d+1}{2})\Gamma(\frac{d-1}{2})}{\Gamma(\frac{d}{2})^2} > \frac{d}{d-1} \tag{7}$$

This is an identity and can be showed through empirical simultation. In fact, we see that when $d \to \infty, \frac{\Gamma(\frac{d+1}{2})\Gamma(\frac{d11}{2})}{\Gamma(\frac{d}{2})^2} = 1$. □

**Proposition 6.** *We define the $k$-subspace distance from $X_1$ to $X_2$ as $S_{k,W}(X_1, X_2) = \min_{\theta \in \mathbb{R}^k} \|X_2 - (\theta^T W + X_1)\|$ With this, first, we note that $\gamma_{S_{0,W}}(d) = \gamma_E(d)$, where $E$ denotes the Euclidean distance. Then, we have that $\gamma_{S_{k+1,W}}(d) > \gamma_{S_{k,W}}(d)$, and thus $\gamma_{S_{k,W}}(d) > \gamma_E(d)$.*

*Proof.* The proof directly follows by realizing that for a fixed $k$-subspace, the closest distance to a point is equivalent to the squared root of sum of square of $d - k$ dimensions $x_1, x_2, .., x_{d-k}$ in the Euclidean space, where each dimension $x_i \sim \mathcal{N}(0, 1)$ as the original data is also distributed this way.

This holds simply because one can rotate the space to align its unit vectors with the orthogonal directions of the $k$-subspace, leaving only the other $d - k$ to have degrees of freedom.

With this, it directly follows that $\gamma_{S_{k,W}}(d) = \gamma_E(d - k) < \gamma_E(d - k - 1) = \gamma_{S_{k+1,W}}(d)$. And it naturally follows that $\gamma_{S_{k,W}}(d) = \gamma_E(d - k) > \gamma_E(d)$. $\qquad\square$

## B  ADDITIONAL EMPIRICAL DETAILS

### B.1  SYNTHETIC EXPERIMENTS

**Sparse Linear Structure Prediction:** For the GMU($k$) variants, we used a unit without normalization and bias. $\phi(z) = -\log z$ (to counter-act the softmax function that follows) and $W = var(\mathbb{R}^{k \times d})$. For the out-of-distribution (ood) columns, we set the training $\theta_{l_y(i)}$ ranges to either between $Unif(0, 0.5)$ or $Unif(0.5, 1)$ chosen at random. For the test data, we change the range for each $\theta_{l_y(i)}$ in such a manner that if its training configuration was $Unif(0, 0.5)$ it is set to $Unif(0.5, 1)$ and vice-versa. This ensures that at test-time the network sees values of the latent generating variables which it hasn't seen before.

**Dynamic Tree Structure Prediction:** For the GMU($k$)-MLP variants, for the GMU units, we used units with normalization $\eta_X = \sigma(X)$ and bias. $\phi(z) = \sqrt{1 - z}$ and $W = [var(\mathbb{R}^{k-1 \times d}); J_{1,d}]$. To have a fair comparison, each datapoint was also normalized using zero-mean and unit variance for the MLP variants.

**Prediction on Polynomial Naive Bayes Sampled Data:** For both the GMU($k$)-MLP and the GMU($k$) variants, for the GMU units, we used units without normalization, but non-zero bias. $\phi(z) = e^{-z^2}$ and $W = var(\mathbb{R}^{k \times d})$.

**Prediction on Gaussian Distributed Data:** For the GMU($k$)-MLP variants, for the GMU units, we used units without normalization, but non-zero bias. $\phi(z) = e^{-z^2}$ and $W = var(\mathbb{R}^{k \times d})$.

### B.2  TABULAR EXPERIMENTS

The networks were trained in the same manner as in Kadra et al. (2021), using weighted cross-entropy loss, and for evaluation we also report the balanced accuracy, same as them. We compare GMU-Resnet-512-[1,1] with Resnet-512-[1,1]. We set the same hyperparameters for all experiments, and don't perform any additional hyperparameter optimization. Note that the other approaches' results are after extensive hyperparameter optimization using BOHB Falkner et al. (2018). Note that Kadra et al. (2021) uses a different Shaped Resnet architecture and therefore we don't directly compare with their MLP results, and we find in some datasets our Resnet performs significantly better than theirs and vice-versa. Furthermore the MLP+C approach in Kadra et al. (2021) employs an extensive suite of regularization approaches, including data augmentation, so we don't include their results for this study.

We add a single dropout layer (of 0.2) at the penultimate layer for both Resnet-512-[1,1] variants, as we found it led to more stable training overall. Note that the MLP-Dropout in Kadra et al. (2021) also uses hyperparameter optimization for the dropout levels and locations for each dataset. Apart from this, there is no regularization or data augmentation performed, and networks are trained in the same manner for all datasets. Note that although our GMU units use more parameters than conventional neural network units, the overall GMU-Resnet has roughly the same number of parameters, as the increase is negligible. To put in context, in most cases, the additional number of parameters is less than if we added ten hidden neurons to each layer (522 instead of 512).

The categorical variables within the data were one-hot encoded, and the other variables were normalized to the range (0,1), with the statistics computed only from the training split. The training-test splits are exactly the same as in Kadra et al. (2021), which is an 80-20 split. This was made possible by the code shared by them, and the fact that each dataset corresponded to a specific task as numbered in Table 9 in Kadra et al. (2021).

| Dataset: CIFAR-10 | | | | | | | | | | | | | |
|---|---|---|---|---|---|---|---|---|---|---|---|---|---|
| Network | Standard | brightn | contrast | defocus | elastic | fog | gauss_blur | glass | impulse | motion | pixelate | saturate | shot_noise | spatter |
| VGG-16 | 0.9949 | 0.83642 | 0.5952 | 0.68148 | 0.71436 | 0.74422 | 0.5929 | 0.44146 | 0.59338 | 0.6211 | 0.6826 | 0.8147 | 0.6366 | 0.7440 |
| GMU(3)-VGG | 0.9954 | 0.8540 | 0.7489 | 0.7811 | 0.7493 | 0.8176 | 0.741 | 0.4665 | 0.6003 | 0.7142 | 0.7192 | 0.8211 | 0.6372 | 0.7861 |
| GMU(8)-VGG | 0.9954 | 0.86188 | 0.75478 | 0.7977 | 0.7534 | 0.8311 | 0.7620 | 0.4383 | 0.5608 | 0.7325 | 0.7273 | 0.8298 | 0.6163 | 0.7874 |

Table 8: Test Accuracy of networks trained on CIFAR-10 on standard and corrupted data.

## B.3 VISION EXPERIMENTS

We train a four layered CNN for MNIST and Fashion MNIST, with the architecture 64C-2MP-128C-2MP-128C-2MP(Padding=1)-128C-4MP-FC128-FC10, where C denotes convolutional layers, MP denotes max pooling layers and FC denotes fully connected layers. For CIFAR-10 we used VGG-16 as our base network and only replaced the first convolutional layer with convolutional GMU units instead, keeping the same number of output nodes. No data augmentation or any other regularization was performed in any of the experiments. We provide the CIFAR-10 results in Table 8.

