# OpenReview forum: "Generative Matching Units for Supervised Learning"
_ICLR.cc/2025/Conference — ICLR 2025 Conference Withdrawn Submission_

### Official Review · Reviewer_7VtZ · 2024-10-31

**Soundness:** 2
**Presentation:** 2
**Contribution:** 2
**Rating:** 3
**Confidence:** 4

**Summary:**

This paper presents a new learning paradigm as an alternative to MLPs. The authors argue that MLPs have an inductive bias suitable for tasks where each datapoint’s label arises from a feedforward process. As opposed to that, they provide certain examples, where the label is involved in the generating process of the datapoint. Motivated by that, they design *Generative Matching Units (GMUs)*, a learning paradigm with an inductive bias towards such tasks. GMUs compute the classification label of a datapoint by solving a minimisation problem that finds the class that has the smallest generation error. To achieve that they define a parametric generative model that is conditioned on the label and train its parameters via classical gradient-based optimisation algorithms. To avoid dealing with a nested optimisation problem, in practice, their generative model is linear, whose optimal value can be found analytically (least squares).

The proposed GMUs are analysed w.r.t. various aspects: First various parameterisations of multi-layer GMUs are discussed and it is shown that they are a generalisation of RBFs. Second, they examine their expressive power, where for the RBF case universality is known, while for the general case, intuitive arguments via a certain construction are given. Third, they introduce the notion of the *information factor*, which is used to broadly argue that GMUs may not suffer from the curse of dimensionality as much as MLPs. Experimentally, the method is tested against several synthetic and real-world experiments showing competitive performance to MLPs, especially in out-of-distribution data.

**Strengths:**

- **Novelty**: The authors introduce an interesting and refreshing idea, challenging the inductive biases of MLPs and showcasing examples where MLPs might have trouble generalising.
- **Evaluation**. The method has been extensively evaluated, showing promising performance in several cases. Also, it is important to note that failure cases of both MLPs and GMUs are identified.

**Weaknesses:**

- **Motivation and Presentation**: Although the authors have attempted to motivate and analyse their method from various perspectives, I must admit that the motivation is still unclear to me. The reason for that may be that there are issues with the presentation (this is why I am discussing both aspects simultaneously). In detail:
    - The paper is hard to follow and its current structure does not facilitate the flow of information.
    - Several sections, although intended to motivate the approach, turned out to be confusing. In particular, “Directed Measure of Functional Dependence” does not seem to provide convincing arguments, while section 4 does not flow naturally from the previous arguments and it is unclear why this is sufficient justification for the curse of dimensionality.
    - Additionally, it is unclear why in the motivating examples of the introduction, one needs a GMU while an MLP is insufficient (or more generally why these motivate the need for a new learning paradigm).

- **Method**: Additionally, there are a few concerns regarding the method:
    - Most design choices seem ad hoc. Also, the fact that the authors try to avoid the inner optimisation loop restricts those choices and it is unclear how this affects the expressive power of the approach.
    - I believe that the theoretical arguments should be improved (this will also help with the motivation part). Currently, only an intuitive argument for universality is provided, while the inductive bias claims are only partially validated experimentally (e.g. the authors could try providing a generalisation bound argument assuming a certain task structure).

- **Experiments.**
    - Even though the experimental evaluation is extensive, it was quite hard to figure out what are the empirical advantages of the proposed approach. In general, the experimental section was a bit hard to follow. For example, the results in all Tables are difficult to interpret and the key takeaways are not adequately highlighted (note that it would be useful to discuss those early on as well, e.g. in the intro). I suggest the authors improve this section substantially and discuss the above in their rebuttal to help the reviewers identify the advantages/disadvantages (generalization? convergence speed?) of the method.
    - Additionally, runtime + training time plots are expected to understand the trade-offs (especially due to the need to solve the least squares problem).

- **Related work**. Finally, a very important concern with this paper is that it completely lacks contextualization. No related work section is provided, which makes it hard to understand how it can be connected to relevant methods.

**Questions:**

- **Minor.**
    - I believe the closed-form solution of the least squares should be included and discussed.
    - Citations should be fixed (they should be written inside parentheses –  use citep instead of cite). The same holds for equations (eqref should be used).

---

### Official Review · Reviewer_vWvJ · 2024-11-01

**Soundness:** 2
**Presentation:** 2
**Contribution:** 2
**Rating:** 5
**Confidence:** 2

**Summary:**

This paper proposes an alternative computational unit, Generative Matching Units (GMUs), for feedforward supervised learning architectures. The generalization ability of GMU is compared with MLPs in comprehensive synthetic experiments and real-data experiments. It is shown in these experiments that GMU-MLPs generalize better than the MLP baselines in most cases. On vision datasets, when compared with the performance of the CNN baseline, GMU-CNNs demonstrate better generalization.

**Strengths:**

1. This paper proposed an alternative computational unit called GMU and its advantage is verified through synthetic and real data experiments.

2. Synthetic data experiments with different types of distribution are conducted, and GMU variants show better generalization, especially to out-of-distribution test data.

**Weaknesses:**

1. The structure of this paper could be clearer to help readers understand it better.  For example, the Introduction section includes the literature review together with the motivation and definition of GMU, and it is followed by an alternative definition of GMU in Section 1.1. Perhaps the clarity of the organization could be improved, and the summary of the organization of the whole paper could be added in the Introduction section.

2. The experiments on real datasets (Section 6) could be explained in more detail. Additionally, the results in Table 6 are not referenced in the paper.

3. There are some typographical errors in this paper, such as ‘mathcalN’ in line 425.

4. The abstract contains many specific equations and notations. The authors could provide a more intuitive overview of this paper here without mentioning too many detailed notations and equations.

**Questions:**

1. In this paper, GMU is compared with MLP. How does the performance of GMU compare with other newly proposed unit structures, for example, those structures mentioned in the Introduction section?

---

### Official Review · Reviewer_Dsi8 · 2024-11-02

**Soundness:** 3
**Presentation:** 3
**Contribution:** 3
**Rating:** 6
**Confidence:** 2

**Summary:**

This paper interprets feedforward computation through a generative perspective, proposing the Generative Matching Units, an alternate computational unit for feedforward supervised learning models. The output of a GMU is a generative error encoding how well the input dimensions of $x$ can be represented as a function of the weights. This can be modeled as a simple least squares problem. An argument is present how GMUs can be universal approximators. GMUs are validated in a variety of synthetic experiments, as well as on multiple tabular and vision datasets. GMUs also show stronger robustness to corruptions.

**Strengths:**

* To the best of my knowledge, the proposed feedforward unit and its generative perspective are novel.
* Numerous experiments are run on synthetic datasets, 30 tabular datasets, and 3 vision datasets, showcasing how GMUs can generalize better than CNNs, MLPs, etc.
* A variety of theoretical remarks are provided, validated by synthetic experiments. For instance, GMUs capture more information in
high-dimensional spaces compared to RBFs and MLPs.

**Weaknesses:**

* The evaluation is in fairly small-scale settings. For instance, the vision experiments are on CIFAR-10, MNIST, Fashion-MNIST.
* The biggest weakness is that there lacks an analysis of scalability. Ultimately the utility of GMUs will be determined by its scaling laws.
* The family of functions that the GMU can use are limited such that the GMU can be formulated as a linear least squares problem for faster computation.

**Questions:**

* How do GMUs do on larger scale datasets and larger scale architectures? Can they be integrated in transformers for both vision and NLP tasks?
* What are the scaling laws for GMUs?
* What are the computational demands and efficiency compared to other architectures?
* A derivation of the gradient flow and updates would be beneficial for the reader to understand the GMU.

---

### Official Review · Reviewer_xDDP · 2024-11-03

**Soundness:** 1
**Presentation:** 1
**Contribution:** 3
**Rating:** 3
**Confidence:** 4

**Summary:**

The Generative Matching Unit is introduced as an alternative building block for neural networks. The GMU uses a generative approach to computing responses to input values. It is based on the idea that the input can be causally dependent on the class label. It could also be seen as a generalisation of the radial basis function. In the manuscript, it is argued that this novel modeling paradigm could lead to better generalisation and overall performance for specific problems, where input and target are related to each other in a causal way. Experiments have been carried out in synthetic datasets, tabular datasets and simple vision datasets.

**Strengths:**

The proposed computation unit is original and is sufficiently distinguished from existing work, it could prove significant for the specific scenarios that it was designed for. The mathematics of the method are sufficient to understand the core assumptions and how it could be further developed. Attempts to prove the unit as capable of universal approximation have been made, and geometric intuitions on what it does have been provided.

**Weaknesses:**

The text is not very clear. The notation is often inconsistent, and the explanations are convoluted. For example, in the paragraph that start from line 080, the weight $W_j'$ is mentioned, which is not used later in the paragraph. This makes it hard to understand what the relationship between $W'_j$ and $W$ is. Then, in line 103, $W_j$ is mentioned again, but now it is together with bias $b_j$. Is $b_j$ a coefficient, as the $b_t$ in equation (1), or is it a full vector?
Another example is notation used in definition 3. Conventionally, $\sim$ is used to described a random variable sampled according to a certain distribution. However, in definition 3 it is used instead of the $\in$ symbol, or at least that is how I interpret it. This is because later in the definition, it is used more correctly by writing $\sim Unif$, which is usually written as $\sim \text{U}$ to indicate that it is sampled from a uniform distribution.
Using bold and other well-established notation conventions, together with consistency in the naming of variables would greatly improve readability.

Overall, the manuscript lacks figures and the ones present are not sufficient. The table is also incredibly hard to parse, requiring a read of the text to understand what is meant with $var$, and being very obscure about the actual meaning of these choices. A lot of space is used for this table, but it is not very useful, it could have been in the appendix, as it is not very relevant for the main text.

The introduction doubles as a method section and there is no related work. I would suggest re-structuring the manuscript, making the introduction more high-level, and providing a more complete context of where GMU places itself when compared to other alternative architectures that are only mentioned, but never explained or properly compared.

The current version of the manuscript does not allow for reproducibility of the results from an expert. This is a major issue. There is no information on how the architecture is actually constructed, or no presudocode on how the training is performed. What is the computation graph, what is the precise closed-form solution to the least-squares problem used, and so many other details that are lacking but essential.

The lack of explanation of the actual architecture used makes it impossible to properly evaluate the results shown. One of the architectures used is described as a single layer of linear GMU units followed by an MLP. Given this description, as long as the GMU can output the identity, the GMU-MLP will also be a universal function approximation, given that MLPs are universal function approximations. The current geometric argument for universal function approximation is rendered very un-intuitive by the absence of a good figure which could clarify the argument. A figure is indeed present in the appendix, but it would be better placed in the main text.

No other functions for $f$ were explored other than a linear relationship. The formulation is generic, at least one experiment with a difference function $f$ should have been present.

The experiments are presented in a very unclear way. There is not bold in some tables, while bold in others. It is not clear what is being bolded, please mention it in the caption. there is very little analysis of the experiments carried out. The synthetic datasets are not exemplified, which makes it hard to understand what they looks like from just the mathematical description.

The core issue is that I do not understand what the architecture is, so I cannot make an assessment on the significance of the results. What is a GMU-ResNet exactly? How did a GMU become convolutional? Please provide pseudocode. As an example of what I do not understand: GMU(k)-MLP-512. This is described as "GMU layer with 512 hidden units followed by a linear layer". How exactly is a GMU layer composed as a computational graph, the 512 hidden units refers to the number of GMU units, or the number of hidden units of the MLP that follows the initial GMU layer? For the ResNet GMU, are there skip-connections around the GMU unit? Is the GMU-ResNet simply a single GMU layer followed by a ResNet?

The limitations are mentioned throughout the text, but should be put into a separate section. It is especially important to highlight how the capacity of the GMU related to the capacity of traditional networks.

**Questions:**

"As most high dimensional data in nature likely has a low dimensional representation due to it existing in low dimensional manifolds" is stated as a fact. However, this is at best an hypothesis, or it would at least require some references to back it up.

In line 115 "tha networks" instead of "than networks".
Equation 5 has wrong parenthesis.

Wikipedia is referenced for the analytical solution to the least-squares problem. Such a reference is completely un-necessary. Please simply add the solution in the appendix or refer to any statistical theory textbook.

---

### Note · Authors · 2024-11-26

I have read and agree with the venue's withdrawal policy on behalf of myself and my co-authors.